# Media consumption and psychological distress among older adults in the United States

**Shawn Bauldry** [ID]*, **Kevin Stainback**

Purdue University, West Lafayette, IN, United States of America

* sbauldry@purdue.edu

## Abstract

The consumption of news media covering national and global events, particularly those that invoke fear or worry, such as pandemics or terrorist attacks, may affect older adults' mental wellbeing. Using the COVID-19 pandemic as a case study, this research analyzes nationally representative data from older adults in the US to address the following research questions: (1) What is the relationship between COVID-19-based media consumption and psychological distress? (2) Does any relationship between media consumption and psychological distress vary by gender, race/ethnicity, education, and marital status? Results indicate that (1) older adults who closely followed the news about the pandemic scored higher on psychological distress than those following less closely and (2) this relationship was more pronounced among Hispanic older adults. These findings are contextualized in the broader stress process model with a focus on a macro-level stressor and differential exposure and vulnerability resulting in variability in the relationship between the stressor and psychological distress.

## Introduction

News media is intended to inform and educate the public about significant national and global events. Research shows that during epidemics, pandemics, and other major events (e.g., terrorist attacks, mass tragedies, or national disasters), people access news more frequently to learn, understand, and reduce uncertainty about events and their implications for their families and communities [1–4]. Despite the benefits of news media consumption for coping with worrisome or difficult events, it also has the potential to promote stress by creating worry and concern for one's health and the health of loved ones. A wide-ranging body of work finds that closely following the news coverage of traumatic events, such as terrorist attacks, natural disasters, or pandemics, is related to increased stress and psychological distress [5–9].

The COVID-19 pandemic provides an important case to analyze the relationship between news media consumption and psychological distress among older adults. The news provided information about the spread of COVID-19, expert recommendations for minimizing the risk of becoming ill, and hospitalization and mortality rates. At the same time, the news also communicated uncertainty and conflicting information from health experts and politicians, stories

**Data Availability Statement:** The data underlying the results presented in the study are available without restriction from https://www.pewresearch.org/our-methods/u-s-surveys/the-american-trends-panel/.

**Funding:** The author(s) received no specific funding for this work.

**Competing interests:** The authors have declared that no competing interests exist.

of personal tragedy, job loss, the dwindling supply of essential goods, shortages of personal protective equipment, and burnout among health care providers. In fact, coverage of the pandemic in the US has been overwhelmingly negative. A recent study found that 91 percent of news stories in the US had a negative tone compared with 54 percent of non-US news stories from major outlets between January 2, 2020 and July 31, 2020 [10].

Recent COVID-19 studies have begun to document the emotional and psychological effects of the pandemic for older adults related to, among other things, loneliness and isolation, familial perceptions of concern, and institutional trust and trust in political leadership [11–15]. To date, however, relatively few studies have examined the relationship between COVID-19 news consumption and wellbeing among adults residing in the US or in other areas of the world [16–19]. Most research has focused on younger populations, though one recent study found a positive effect of COVID-19 media dependency on anxiety for both women and men [20].

The stress process paradigm provides a framework to hypothesize about relationships between media consumption and mental health. Past work, though not featuring the stress process paradigm, primarily emphasizes the direct relationship between media consumption and mental health. A key insight of the stress process paradigm, however, is that exposure and vulnerability to stressors is unequally distributed across populations. We are unaware of any research that systematically examines whether media consumption has differential effects on psychological distress for different subpopulations. As such, our study examines two research questions: (1) What is the relationship between COVID-based media consumption and psychological distress? (2) Does any relationship between news consumption and psychological distress vary by gender, race/ethnicity, education, and marital status?

Our analysis draws on data from Wave 64 of the Pew Research Center's (Pew) American Trends Panel (ATP), a nationally representative sample that includes over 2,500 adults ages 65 and older. Wave 64 of the ATP was fielded between March 19 and March 24, 2020, a period of uncertainty and a rapidly evolving understanding of the pandemic as well as increased news media consumption [21]. The timing of the ATP during the early stages of the pandemic makes it an ideal data source for examining the relationship between media consumption and psychological distress among older adults.

## News media consumption and the stress process paradigm

As of June 30, 2022, COVID-19 is responsible for over 1,000,000 confirmed deaths in the US and over 6,000,000 deaths worldwide [22, 23]. The physical toll of the pandemic has been especially felt among older adults with adults ages 65 and older accounting for 74 percent of the COVID-attributed deaths in the US and a clear age-graded mortality risk [22]. In addition to the far-reaching effects on the physical health of older adults, the pandemic has the potential for an even broader impact on their mental health.

The stress process paradigm, one of the most widely used perspectives in examining how social conditions can influence mental health [24, 25], provides a framework for analyzing the relationship between news media consumption surrounding the pandemic and psychological distress. The stress process paradigm captures the connections between various forms of stressors, psychological distress, and different types of psychosocial resources that may buffer the effects of stressors. Past studies have primarily examined the impact of micro-level stressors, such as experiences of discrimination or the loss of a partner, but stressors can operate at the meso-level (e.g., neighborhood disorganization) and macro-level (e.g., an economic recession) as well [26].

For this study, we view the COVID-19 pandemic as a macro-level stressor that has the potential to be a significant source of stress for older adults in the US and around the world. It

is a global event that has disrupted social life in every regard. Beyond its direct consequences for illness, hospitalization, and death, it has altered social routines leading to isolation and loneliness, and increased burdens on older adults and their families with potential job loss and managing children or grandchildren unable to attend school or daycare. The pandemic generated enormous uncertainty about the future, about a vaccine, and about the health of oneself and loved ones.

Furthermore, news media acts as a conduit of this macro-level stressor, such that the stressor is more likely to be activated for older adults who closely follow the news. Even for those not directly affected by COVID-19, following the news translates the pandemic into an anticipatory stressor, one in which learning about the experiences of others can lead individuals to consider their own, their families', and their extended network's susceptibility to the stressor [27]. This leads us to our first hypothesis

> H1: *Older adults who follow the news media surrounding the pandemic more closely will exhibit greater psychological distress than those who follow the news media less closely.*

One of the core ideas of the stress process paradigm is that the exposure to stressors and the availability of psychosocial resources to buffer their impacts are not evenly distributed in the population. On average, older adults in disadvantaged social positions are more likely to be exposed to the negative health and social impacts of the pandemic and less likely to have a reserve of psychosocial resources to address these impacts. Once again, viewing closely following the news as a conduit of the stressors of the pandemic, we expect the relationship between closely following the news and psychological distress to vary depending on social position. In our analysis we focus on three of the most salient dimensions of social position among older adults living in the US–gender, race-ethnicity, and socioeconomic resources.

Around the world, the COVID-19 pandemic has had an unequal impact for women and men. Although men appear to be at a slightly greater risk for adverse health outcomes and mortality from infection, women have experienced more significant social impacts with greater effects on, for instance, employment and caregiving [28, 29]. In the US racial-ethnic minorities, particularly Hispanic and Black people experienced greater exposure to COVID-19 and suffered worse health outcomes than White people [30]. Finally, people with higher levels of education, particularly college degrees, experienced fewer disruptions from the pandemic and were better able to manage a transition to remote work than people with lower levels of education [31]. These patterns lead to our next three hypotheses:

> H2: *The relationship between closely following the news media surrounding the pandemic and psychological distress will be greater for women than for men.*

> H3: *The relationship between closely following the news media surrounding the pandemic and psychological distress will be greater for racial-ethnic minorities than for White older adults.*

> H4: *The relationship between closely following the news media surrounding the pandemic and psychological distress will be greater for older adults with lower levels of education than those with higher levels of education.*

Social support is among the most studied psychosocial resource that can alleviate the effect of stressors [27]. Among older adults marital relationships are a particularly important form of social support [32, 33]. As such, we would expect that any anticipatory stressor activated from

closely following the news media surrounding the pandemic may be dampened by the presence of a spouse or partner providing support. This leads us to our final hypothesis.

> H5: *The relationship between closely following the news media surrounding the pandemic and psychological distress will be greater for older adults who are living with a spouse or partner than those who are not.*

In sum, based on the stress process paradigm and viewing the COVID-19 pandemic as a macro-level stressor and the media as a conduit, we expect to see a relationship between closely following the media surrounding the pandemic and psychological distress on average as well as variation in the relationship for specific subgroups of the population.

### Psychological distress and media consumption

The stress process paradigm emphasizes the effects of stressors on psychological distress, but it is also possible that psychological distress may lead to exposure to additional stressors, a reciprocal process involving reverse causation. In the context of this study, for instance, people experiencing higher levels of psychological distress may be more likely to follow the news surrounding the pandemic. Few studies in this area have investigated the possibility that psychological distress can lead to more news media consumption. One study, however, found evidence of a cyclical process between media consumption and acute stress following the 2013 Boston Marathon bombings and the 2016 Orlando Pulse nightclub massacre [34]. We believe this type of reciprocal and amplifying process in which more closely following the news can lead to psychological distress, which in turn leads to more media consumption, is likely for the COVID-19 pandemic as well, an event with a much longer duration than terrorist attacks or mass killings. To help account for this possibility, we adjust for having a prior mental health condition in the following analyses, but we return to a consideration of this issue in our discussion.

## Data and methods

Our analysis draws on data from Pew's American Trends Panel (ATP). The ATP is an online, ongoing, nationally-representative, probability-based sample of non-institutionalized adults, ages 18 and older, residing in the United States [35]. The data for our analysis come from Wave 64, the first wave to include questions specifically about media consumption related to COVID-19 and psychological distress. A total of 11,537 respondents participated in this wave of data collection between March 19 and March 24, 2020. The survey was conducted in both English and Spanish and the response rate for this wave was 78.4 percent. Our analysis sample is restricted to respondents 65 and older who identified as either Hispanic, non-Hispanic Black, or non-Hispanic White (N = 2,843). We dropped a small number of cases (N = 62) missing data for any of the covariates, which resulted in an analysis sample of N = 2,781. The data for this analysis is publicly available and can be downloaded at www.pewresearch.org/american-trends-panel-datasets/.

### Measures

Our outcome is a scale constructed from taking the sum of five measures of psychological distress adapted from the Center for Epidemiologic Studies Depression [36] and the Generalized Anxiety Disorder [37] scales and have been used in a number of past studies as a broad measure of psychological distress [9, 38, 39]. The measures, share the stem "[i]n the past 7 days, how often have you . . ." (a) felt nervous, anxious, or on edge, (b) felt depressed, (c) felt lonely,

(d), felt hopeful about the future, and (e) had trouble sleeping (see Table A1 in S1 Appendix for descriptive statistics). Responses ranged from (1) "rarely or none of the time (less than 1 day)" to (4) most or all of the time (5–7 days)." We reverse-coded the fourth item before summing the measures. We estimate a reliability for the scale of 0.74 based on a version of coefficient omega that relies on fewer assumptions than Cronbach's alpha and accounts for the ordinal measurement of the indicators [40].

Our focal independent variable is media consumption related to COVID-19 as measured by the question "[h]ow closely have you been following the news about the outbreak?" with response categories "not at all closely," "not too closely," "fairly closely," and "very closely." The two lowest response categories accounted for only two percent of cases, thus we collapsed the measure into an indicator for "very closely" versus the remaining three categories.

Based on hypotheses 2 through 5, our focal moderators include gender, race/ethnicity, educational attainment, and marital or relationship status. Our measure of gender is a binary indicator for women, our measure of race/ethnicity includes three categories for Hispanic, non-Hispanic Black, and non-Hispanic White older adults. Our measure of educational attainment includes three categories for completing a high school degree or less, completing some college, and completing a college degree or more. Our final moderator is an indicator for being married or living with a partner versus those who reported being divorced, separated, widowed, or never married.

In addition to our focal variables, we adjust for factors potentially related to both COVID-19 media consumption and psychological distress. Given that the spread of the pandemic in the early stages was particularly concentrated in urban areas in the Northeast and the West, we include an indicator for residing in a metropolitan area and a series of indicators that capture four major regions (Northeast, South, Midwest, and West) in our models. In addition, as discussed above, people with higher levels of psychological distress may be more likely to follow the news closely. To help account for this, we include an indicator for respondents who have ever had a doctor or health care provider tell them that they have a mental health condition.

### Analysis strategy

Our analysis proceeds in three steps. First, we examine descriptive statistics stratified by COVID-19 news media consumption to document bivariate differences in psychological distress as well as sociodemographic correlates of news media consumption. Second, we assess our first hypothesis by regressing psychological distress on news media consumption and all other covariates. Third, we assess our four hypotheses concerning variation in the relationship between media consumption and psychological distress by fitting separate models for each subgroup defined by our candidate moderators (e.g., separate models for women and men). This strategy is equivalent to introducing interaction terms between the candidate moderator and all variables in the model and permits the control variables to have different associations with psychological distress across subgroups. We then test for differences in the relationship between media consumption and psychological distress across models using standard difference tests [41]. We use ATP sample weights to adjust for unequal probabilities of sample selection for all descriptive statistics and models. All analyses were conducted in R and code for data preparation and analyses is deposited on OSF to facilitate replication and extensions (https://osf.io/ndqbv/).

### Results

Table 1 reports weighted descriptive statistics for our sample overall and stratified by COVID-19 news media consumption. We see that about 75 percent of respondents closely followed the

**Table 1. Descriptive statistics overall and by COVID-19 media consumption.**

|  | Overall | Media consumption | | Difference |
|---|---|---|---|---|
|  |  | VC | Not VC |  |
|  | (N = 2,781) | (N = 2,066) | (N = 715) | VC vs not VC |
|  | Mean | Mean | Mean |  |
| Psychological distress | 9.29 | 9.47 | 8.88 | 0.59*** |
| Women | 0.55 | 0.57 | 0.50 | 0.07*** |
| Hispanic | 0.05 | 0.06 | 0.04 | 0.02* |
| non-Hispanic Black | 0.07 | 0.07 | 0.08 | -0.01 |
| non-Hispanic White | 0.88 | 0.88 | 0.88 | 0.00 |
| High school or less | 0.37 | 0.33 | 0.46 | -0.13*** |
| Some college | 0.31 | 0.32 | 0.28 | 0.04* |
| College degree or more | 0.32 | 0.35 | 0.26 | 0.09*** |
| Married/partner | 0.59 | 0.61 | 0.55 | 0.06** |
| Metropolitan area | 0.84 | 0.84 | 0.85 | -0.01 |
| Northeast | 0.17 | 0.19 | 0.12 | 0.07*** |
| Midwest | 0.24 | 0.22 | 0.28 | -0.06*** |
| South | 0.37 | 0.37 | 0.38 | -0.01 |
| West | 0.22 | 0.22 | 0.23 | -0.01 |
| Prior mental health condition | 0.09 | 0.09 | 0.09 | 0.00 |

*Notes*: *$p < 0.05$,

**$p < 0.01$,

*** $p < 0.001$. Weighted descriptive statistics. VC refers to very close media consumption. Statistical significance for differences assessed with t-tests.

news surrounding the pandemic. We observe a difference of 0.59 in average psychological distress between older adults who reported following COVID-19 news media very closely and those who did not. This is not a particularly large difference in psychological distress, about a sixth of a standard deviation, but it is notable given the early stage of the pandemic. We also note that women, older adults with higher levels of education, and older adults living in the northeast were all more likely to report following COVID-related news stories more closely than their respective counterparts. Notably we do not observe bivariate differences in media consumption by race-ethnicity, metropolitan area, or having a prior mental health condition.

Fig 1 reports estimates of coefficients from regressing psychological distress on media consumption and all other covariates (also see Table A2 in S1 Appendix). Older adults who followed COVID-19 news very closely score 0.63 higher on the psychological distress scale than older adults who did not follow the news as closely after adjusting for other factors, including prior mental health conditions. Although the magnitude is not large, this is roughly equivalent to the net gap between women and men, 0.64, in psychological distress, an often-studied mental health disparity. This provides support for Hypothesis 1 that on average closely following the media surrounding the pandemic has a relationship with psychological distress.

To assess Hypotheses 2 through 5, we fit separate models by gender, race-ethnicity, education, and current marital/cohabitation status, respectively, and examine estimates of the relationship between very close media consumption, net of all other covariates, and psychological distress as well as predicted values of psychological distress. Fig 2 illustrates estimates of very close media consumption for the various subgroups indicated on the y-axis (Panel A) and predicted values of psychological distress (Panel B) (also see Table A3 in S1 Appendix). Beginning with gender, we find a greater association between very close media consumption and psychological distress for women than for men, but there is a high degree of overlap in the confidence

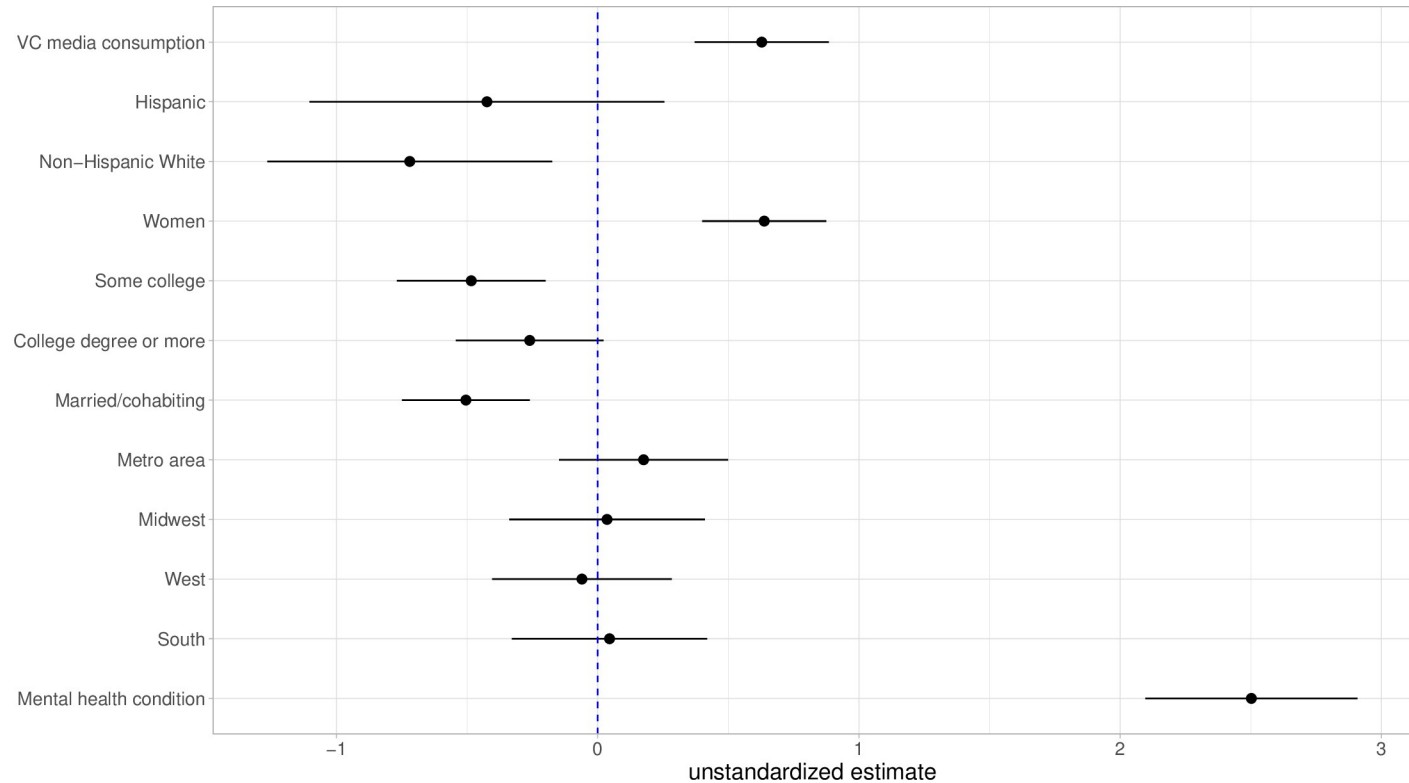

**Fig 1. Estimates of coefficients from regressing psychological distress on media consumption and all other covariates.** Estimates are unstandardized with 95 percent confidence intervals. See text for the referent categories. The regression model incorporated the sample weights.

intervals and the difference between the two estimates is not statistically significant. For both women and men, closely following the news media surrounding the pandemic is related to higher levels of psychological distress. We observe a similar pattern when comparing the relationship between very close media consumption and psychological distress among older adults who were married or living with a partner as compared with those living alone. We do not find support for Hypothesis 2 or Hypothesis 5.

Turning to race-ethnicity, we find evidence of different relationships between closely following the news media and psychological distress across subgroups. We observed a much stronger relationship among Hispanic older adults than either non-Hispanic Black or non-Hispanic White older adults. In terms of predicted values, Hispanic older adults who closely follow the news media surrounding the pandemic have the highest levels of psychological distress of any subgroup. We also observe a higher estimate for very close media consumption among Black older adults compared to White older adults, but it comes with a particularly large confidence interval that renders the difference non-significant. These patterns provide support for Hypothesis 3 with respect to Hispanic older adults.

Finally, we observe a clear gradient, albeit not statistically significant, across education levels such that closely following the news media has a stronger relationship with psychological distress for older adults with higher levels of education. In fact, there is little evidence of a relationship between media consumption and psychological distress at all for older adults with a high school degree or less. Notably, older adults with a college degree or higher who follow the news media closely have above average predicted psychological distress, while those who do not follow the media closely have among the lowest levels of predicted psychological distress.

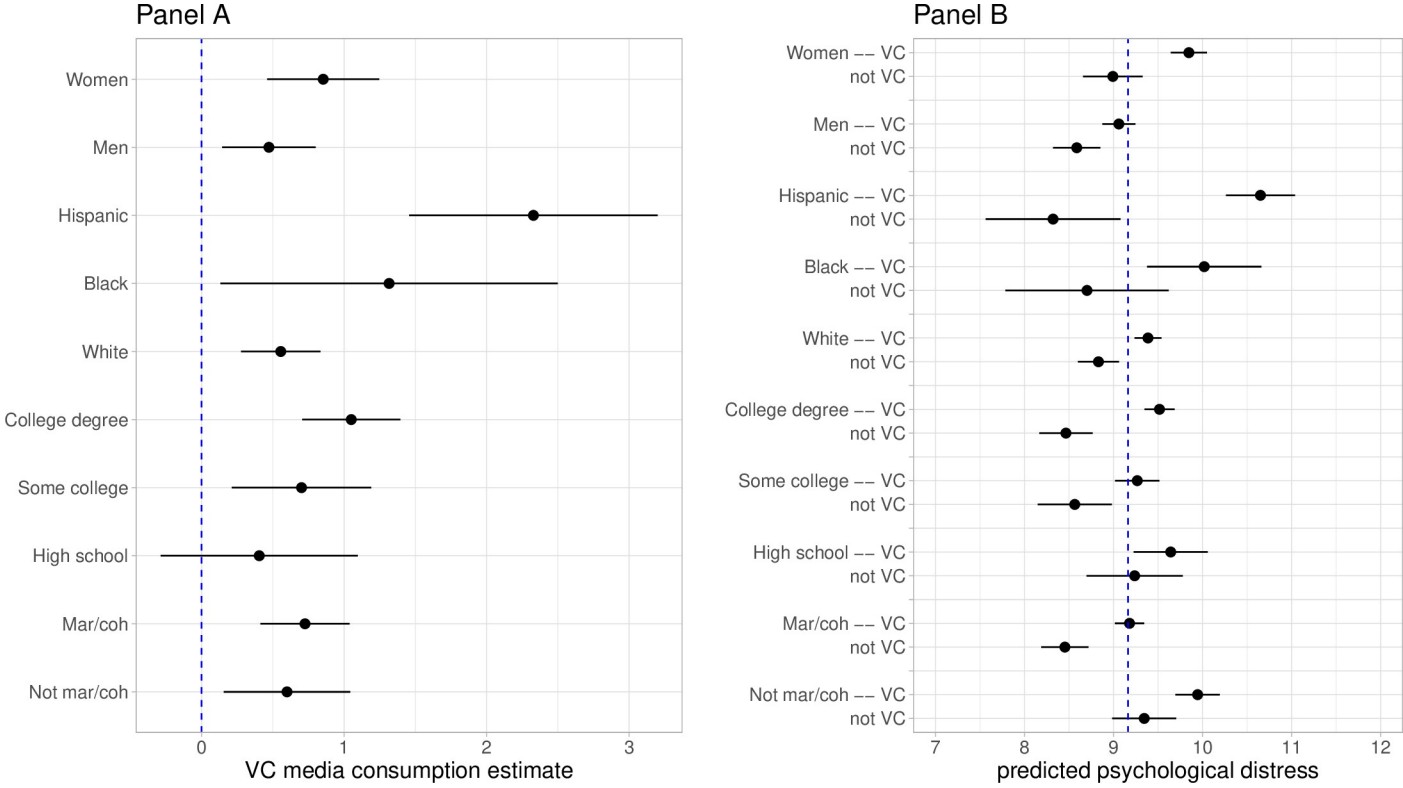

**Fig 2.** Estimates of coefficients (Panel A) and predicted values (Panel B) from regressing psychological distress on very close media consumption stratified by subsamples indicated on the y-axis. Estimates are unstandardized with 95 percent confidence intervals. All models are weighted and adjust for all covariates outlined in the text. VC = very close. Not VC = not very close. The dashed line in Panel A marks an estimate of 0 and the dashed line in Panel B marks the weighted average level of psychological distress among all respondents.

This pattern is in the opposite direction of Hypothesis 4, as we expected a stronger relationship for older adults with lower levels of education due to greater vulnerability to the pandemic and fewer psychosocial resources on average. We return to consider the implications of this in our discussion.

## Discussion

This works joins a growing body of research documenting the potential negative effects of closely following the news media surrounding national or global traumatic events on mental health. For our analysis, we focused on older adults, adopted the stress process paradigm as a theoretical framework, and viewed the COVID-19 pandemic as a macro-level stressor that was more strongly activated by closely following the news. In support of our first hypothesis, we found evidence a relationship between closely following the news surrounding the pandemic and psychological distress with a similar magnitude to the gender gap in psychological distress among older adults.

Drawing on a key insight of the stress process paradigm that exposure and vulnerability to stressors as well as psychosocial resources to buffer the effects are unevenly distributed, we further hypothesized that older adults in disadvantaged social positions and older adults living without a spouse or partner would be more affected from closely following the news surrounding the pandemic. For these hypotheses, we found mixed support. We do not find evidence of a differential relationship by gender, a finding that is consistent with a past study [20]. We also

do not find evidence of a differential relationship from living with a spouse or partner. This may reflect a difference in how macro-level (or meso-level) as opposed to micro-level stressors operate. It is possible that social support from marriage or cohabitation is less effective as a buffer of stress when the household is subject to the same stressor. In this case, it is likely that if one partner closely follows the news surrounding the pandemic, then the other partner is likely to as well or at least be exposed to the news via their partner. Additional research is needed with direct measures of the stressor and buffering mechanisms to assess these possibilities.

We do, however, find evidence of differential relationships between closely following the news surrounding the pandemic and psychological distress by race-ethnicity. For race-ethnicity, the pattern is consistent with our hypothesis. Particularly in the early stages of the pandemic, racial-ethnic minorities, and Hispanic people in particular, had greater exposure and vulnerability to COVID-19 infection and this likely led to increased concern among Hispanic older adults closely following the news [42]. We also observed a surprising gradient across levels of education, though it was not statistically significant at conventional levels. The pattern in relationships we observed is not consistent with our hypothesis in that our results suggest a potentially stronger relationship for older adults with higher levels of education. This finding is surprising from the perspective of the stress process paradigm. It may in part reflect a partisan leaning with higher educated older adults tending to be Democrats with a greater concern in general about the pandemic. If so, this would represent a departure from the stress process paradigm's focus on exposure, vulnerability, and psychosocial resources and a shift to a broader cultural component. This represents an interesting avenue for future theoretical innovation and empirical investigation.

This analysis has limitations that are important to mention. First, as noted above, we do not have direct measures of the mechanisms of the stress process paradigm. As we have discussed, the overall pattern of results is consistent with the stress process paradigm for a couple of hypotheses but inconsistent with others. Additional research is needed to understand the mechanisms underlying the stress process with a macro-level stressor. Second, we view our analysis as descriptive and broadly consistent with our theoretical framework, but not a causal analysis. Our ability to adjust for prior mental health conditions helps address the potential reciprocal process in which psychological distress may lead people to follow the news more closely, but, of course, it does not fully capture the process and potential amplification in the early stages of the pandemic. Identifying causal effects in one direction or another of such a reciprocal process is likely to be quite challenging, but it remains an important avenue for future work. In the meantime, we believe documenting the relationship is valuable as it points to an underappreciated mechanism in which a traumatic event, such as the pandemic, can shape the mental wellbeing of older adults.

## Conclusion

Using the COVID-19 pandemic as an important case, this study documented a relationship between closely following the news media surrounding the pandemic and psychological distress among older adults. This represents one mechanism through which large-scale traumatic events can affect older adult's mental health. Although the news media provides important information about traumatic events, such as the pandemic, and therefore should not be restricted, an awareness of the potential negative effects can help guide efforts to support the wellbeing of older adults, even those not directly affected.

Furthermore, this study focused on the media as a conduit for a macro-level stressor. Most past research has attended to micro-level stressors and, as a consequence, may have overlooked some of the unique dynamics with respect to exposure, vulnerability, and access to

psychosocial resources that differ when examining the impacts of macro-level stressors. This is an important avenue for future research on the mental health of older adults.

## Supporting information

**S1 Appendix. Contains all of the supporting tables.**
(DOCX)

## Author Contributions

**Conceptualization:** Shawn Bauldry, Kevin Stainback.

**Data curation:** Shawn Bauldry, Kevin Stainback.

**Formal analysis:** Shawn Bauldry.

**Visualization:** Shawn Bauldry.

**Writing – original draft:** Shawn Bauldry, Kevin Stainback.

**Writing – review & editing:** Shawn Bauldry, Kevin Stainback.

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
