## [Decision Letter · Decision Letter 0]

9 Nov 2022

PONE-D-22-26923Media Consumption and Psychological Distress Among Older AdultsPLOS ONE

Dear Dr. Bauldry,

Thank you for submitting your manuscript to PLOS ONE. After careful consideration, we feel that it has merit but does not fully meet PLOS ONE’s publication criteria as it currently stands. Therefore, we invite you to submit a revised version of the manuscript that addresses the points raised during the review process.

We look forward to receiving your revised manuscript.

Kind regards,

Michio Murakami

Academic Editor

PLOS ONE

Journal Requirements:

Reviewers' comments:

Reviewer's Responses to Questions

**Comments to the Author**

1. Is the manuscript technically sound, and do the data support the conclusions?

Reviewer #1: Partly

Reviewer #2: Partly

2. Has the statistical analysis been performed appropriately and rigorously? 

Reviewer #1: No

Reviewer #2: No

3. Have the authors made all data underlying the findings in their manuscript fully available?

Reviewer #1: No

Reviewer #2: Yes

4. Is the manuscript presented in an intelligible fashion and written in standard English?

Reviewer #1: Yes

Reviewer #2: Yes

5. Review Comments to the Author

Reviewer #1: Interesting paper on an important topic. However, I have major concerns about the analytic strategy and conclusions that need to be addressed:

1) Please justify the use of "adapted" definition of GAD scale which was used for the primary outcome. Exactly what was adapted?

2) The main hypothesis of association of increased media exposure with more distress did not stand correct in the unadjusted or even bivariate analysis. Much of the conclusion is based on multivariate model which is fine. However, I would like to see a sensitivity analysis to confirm this is not just a data fluke. Could you do a logistic regression analysis using top 25th percentile of GAD score as an outcome?

3) In the observational study it is entirely plausible that those more prone to anxiety are reading more? Please state potential bidirectionality of the association as a possibility in the Discussion.

Reviewer #2: The title of the research was - Media Consumption and Psychological Distress Among Older Adults and by reading this title the audience would think that the results and the paper would represent the whole globe whereas it represented US only. The authors also have claimed that there were fewer research been made on examining the relationship between COVID-19 news consumption and wellbeing among older adults residing in the US - however there were some good quality papers available which already have explained such issues in depth though it is true that the number of research were few. In addition, the statistical analysis in the paper was not enough to support the hypothesis or the statements made. Moreover the outcomes of the paper are quite well known to the greater audience all through the world and I think such paper will not make a big impact to research world any more.

6. PLOS authors have the option to publish the peer review history of their article (what does this mean?). If published, this will include your full peer review and any attached files.

Reviewer #1: No

Reviewer #2: **Yes: **Dewan Muhammad Nur -A Yazdani

---

## [Author Response · Author response to Decision Letter 0]

22 Nov 2022

Reviewer 1

1. Please justify the use of "adapted" definition of GAD scale which was used for the primary outcome. Exactly what was adapted?

We appreciate the reviewer’s request for more information about the items used in our measure of psychological distress. The measures come from a combination of one item from the GAD-7 scale (Spitzer et al. 2006) and four items from the CESD-20 scale (Radloff 1977). This set of items has been used in previous studies as a broad measure of psychological distress, which is how we view the items (see, e.g., Cobb et al. 2021; Hearne 2021; Stainback et al. 2020). We have revised our discussion of the measures to be more clear about their origins (see p. 9), we include the specific items in our discussion (see p. 9-10), and we added a table with descriptive statistics for the specific items and their origins to our Appendix (see p. 23).

2. The main hypothesis of association of increased media exposure with more distress did not stand correct in the unadjusted or even bivariate analysis. Much of the conclusion is based on multivariate model which is fine. However, I would like to see a sensitivity analysis to confirm this is not just a data fluke. Could you do a logistic regression analysis using top 25th percentile of GAD score as an outcome?

We are uncertain about the reviewer’s concern as there is a bivariate difference in psychological distress between older adults who reported very close versus not very close media consumption surrounding the pandemic. We added bivariate significance tests to Table 1 and discuss the bivariate differences in the first section of our results (see p. 12-13). We appreciate the reviewer’s suggestion for a sensitivity analysis and did indeed find a relationship between media consumption and psychological distress at the 25th percentile (adjusted OR = 1.416; p-value = 0.0006) net of the same covariate we use in our primary analysis. In other words, very closely following the news surrounding the pandemic is associated with a 42 percent greater odds of being in the top quartile of psychological distress relative to not very closely following the news net of all other covariates.

3. In the observational study it is entirely plausible that those more prone to anxiety are reading more? Please state potential bidirectionality of the association as a possibility in the Discussion.

We agree with the reviewer and indicate the potential for bidirectionality in our theoretical model in a section titled “Psychological Distress and Media Consumption” (see p. 8-9), in our decision to include a self-reported past mental health condition as a covariate (see p. 11), and in our discussion where we acknowledge that we have not fully addressed the possibility (see p. 17).

Reviewer 2

1. The title of the research was - Media Consumption and Psychological Distress Among Older Adults and by reading this title the audience would think that the results and the paper would represent the whole globe whereas it represented US only. 

The reviewer raises a good point about expectations. In the prior draft, we used the language of “older US adults” throughout including in the abstract. We have now revised the title of our manuscript to reference older adults in the United States to create consistency between title and text.

2. The authors also have claimed that there were fewer research been made on examining the relationship between COVID-19 news consumption and wellbeing among older adults residing in the US - however there were some good quality papers available which already have explained such issues in depth though it is true that the number of research were few. 

The reviewer is correct that there are a few studies that have examined COVID specific media consumption and psychological distress, which we cite in our paper (see p. 4). We conducted an extensive search to determine if we missed any studies since we drafted our original manuscript and found four studies published this year or late last year that explore COVID media consumption and mental health in different areas of the world. We have added citations to these new studies in our manuscript. To our knowledge, however, no past studies of COVID media and distress have focused on older adults, the population most affected by the pandemic, nor have any past studies examined how the relationship between media consumption and psychological distress varies across theoretically informed subpopulations. These two advances represent significant contributions to the research in this area, which we discuss in our introduction (see p. 4) and highlight in our discussion and conclusion (see p. 15-18).

3. In addition, the statistical analysis in the paper was not enough to support the hypothesis or the statements made. 

We disagree with the reviewer on this point. Our statistical analyses demonstrate two things: (1) there is an association between COVID media consumption and psychological distress net of a variety of sociodemographic factors related to exposure to the pandemic and past mental health conditions (see p. 13; also see note above responding to a comment from Reviewer 1 that the association holds as well for the top quartile of psychological distress) and (2) the relationship is significantly different across racial-ethnic groups along the lines we hypothesized (see p. 13-14). 

4. Moreover the outcomes of the paper are quite well known to the greater audience all through the world and I think such paper will not make a big impact to research world any more.

As noted above, the focus on older adults and variation in the relationship between media consumption surrounding the pandemic and psychological distress represent significant contributions to research in this area. More generally, this study contributes to the area of research on how media consumption surrounding significant traumatic events (e.g., terrorist attacks and natural disasters) can have a broader impact on population mental health. We highlight this contribution as well in our discussion and conclusion (see p. 17-18).

---

## [Decision Letter · Decision Letter 1]

12 Dec 2022

Media Consumption and Psychological Distress Among Older Adults in the United States

PONE-D-22-26923R1

Dear Dr. Bauldry,

We’re pleased to inform you that your manuscript has been judged scientifically suitable for publication and will be formally accepted for publication once it meets all outstanding technical requirements.

Kind regards,

Michio Murakami

Academic Editor

PLOS ONE

Additional Editor Comments (optional):

Reviewers' comments:

Reviewer's Responses to Questions

**Comments to the Author**

1. If the authors have adequately addressed your comments raised in a previous round of review and you feel that this manuscript is now acceptable for publication, you may indicate that here to bypass the “Comments to the Author” section, enter your conflict of interest statement in the “Confidential to Editor” section, and submit your "Accept" recommendation.

Reviewer #1: All comments have been addressed

Reviewer #2: All comments have been addressed

2. Is the manuscript technically sound, and do the data support the conclusions?

Reviewer #1: Yes

Reviewer #2: Partly

3. Has the statistical analysis been performed appropriately and rigorously? 

Reviewer #1: Yes

Reviewer #2: Yes

4. Have the authors made all data underlying the findings in their manuscript fully available?

Reviewer #1: No

Reviewer #2: Yes

5. Is the manuscript presented in an intelligible fashion and written in standard English?

Reviewer #1: Yes

Reviewer #2: Yes

6. Review Comments to the Author

Reviewer #1: Interesting topic. The response to my queries is satisfactory in the revisions. No additional concerns.

Reviewer #2: The authors have adequately addressed my comments raised in a previous round of review and this manuscript is now acceptable for publication. The manuscript have described a technically sound piece of scientific research with data that supports the conclusions. Experiments have also been conducted rigorously, with appropriate controls, replication, and sample sizes. Moreover, the conclusions was drawn appropriately based on the data presented.

7. PLOS authors have the option to publish the peer review history of their article (what does this mean?). If published, this will include your full peer review and any attached files.

Reviewer #1: No

Reviewer #2: **Yes: **Dewan Muhammad Nur -A Yazdani

---

## [Editor Report · Acceptance letter]

19 Dec 2022

PONE-D-22-26923R1 

Media Consumption and Psychological Distress Among Older Adults in the United States 

Dear Dr. Bauldry:

I'm pleased to inform you that your manuscript has been deemed suitable for publication in PLOS ONE. Congratulations! Your manuscript is now with our production department. 

Kind regards, 

on behalf of

Dr. Michio Murakami 

Academic Editor

PLOS ONE